# DSR: Optimization of Performance Lower Bound for Hierarchical Policy with Dynamical Skill Refinement

## Abstract

Skill-based reinforcement learning (Skill-based RL) is an efficient paradigm for solving sparse-reward tasks by extracting skills from demonstration datasets and learning high-level policy which selects skills. Because each selected skill by high-level policy is executed for multiple consecutive timesteps, the high-level policy is essentially learned in a temporally abstract Markov decision process (TA-MDP) built on the skills, which shortens the task horizon and reduces the exploration cost. However, these skills are usually sub-optimal because of the potential low quality and low coverage of the datasets, which causes the sub-optimal performance in the downstream task. It is intuitive to refine the skills. However, it is a hard issue to refine the skills while ensuring performance improvement and avoiding non-stationarity of transition dynamics caused by skill changes. To address the dilemma of sub-optimality and ineffectiveness, we propose a unified optimization objective for the entire hierarchical policy. We theoretically prove that the unified optimization objective guarantees the performance improvement in TA-MDP, and that optimizing the performance in TA-MDP is equivalent to optimizing the performance lower bound of the entire hierarchical policy in original MDP. Furthermore, in order to overcome the phenomenon of skill space collapse, we propose the dynamical skill refinement (DSR) mechanism which names our method. The experiment results empirically validate the effectiveness of our method, and show the advantages over the state-of-the-art (SOTA) methods.

## 1 Introduction

Reinforcement learning has been widely used in complex application scenarios, such as robotics manipulation Gupta et al. (2019); Chane-Sane et al. (2021); Kipf et al. (2019); Nair et al. (2020); Pertsch et al. (2021); Shi et al. (2022); Huang et al. (2023); Pertsch et al. (2022). The tasks in these domains are usually with sparse reward functions which give a positive feedback only when the task is completed. In order to reduce the extremely high exploration costs caused by sparse reward functions, prior works have proposed a variety of methods to incorporate prior knowledge into reinforcement learning Nair et al. (2020); Rajeswaran et al. (2017); Daoudi et al. (2023); Zhang et al. (2023); Wen et al. (2023); Guo et al. (2023); Pertsch et al. (2021; 2022). In these works, skill-based reinforcement learning (Skill-based RL) has become an efficient approach to solve the sparse-reward tasks Kipf et al. (2019); Pertsch et al. (2021; 2022). Skills are temporally extended behaviors and usually manifest as the segments of consecutive actions in demonstration datasets. These skills in the dataset are typically embedded into the latent space of the low-level policy which serves as the action space of the high-level policy. The high-level policy selects the appropriate skills according to the states, and lets each selected skill be executed for multiple consecutive timesteps so as to shorten the task horizon and reduce the exploration cost.

These skills may be sub-optimal because of the potential low quality and low coverage of dataset Fu et al. (2020), which causes the potential sub-optimal performance in downstream task. It is intuitive to refine the extracted skills with online collected transitions while learning the high-level policy. However, how to refine skills while achieving performance improvement is still an unresolved issue. Furthermore, the change of skills will lead to the non-stationarity of the transition dynamics of the temporally abstract Markov decision process (TA-MDP), which we name temporal abstraction shift.

Temporal abstraction shift hinders us from learning the high-level policy because the actual value of any state-skill keeps changing, making it difficult for us to accurately estimate it.

Therefore, we usually have to face either the sub-optimal skills or the ineffective skill refinement, which constitutes a dilemma. Some prior methods assume that skills are nearly optimal and keep them fixed, but this assumption puts high requirements on datasets, which can hardly be met when the downstream task is different from all the tasks used to generate the datasets Kipf et al. (2019); Pertsch et al. (2021; 2022). ReSkill Rana et al. (2023) updates both the high-level policy and the skills in an on-policy RL manner Schulman et al. (2015; 2017), which is an ingenious way to circumvent the temporal abstraction shift. However, in ReSkill, the high-level policy and the skills are updated in the TA-MDP and the original MDP respectively. Inconsistent optimization objectives can not theoretically guarantee the performance improvement. Skill-Critic Hao et al. (2024) directly ignores the temporal abstraction shift and updates both the high-level policy and the skills in the off-policy RL manner Haarnoja et al. (2018b). Skill-Critic achieves superior performance and sample efficiency in some specific tasks. However, ignoring the temporal abstraction shift brings in uncertainty, and Skill-Critic depends on SPiRL-based Pertsch et al. (2021) warm-up stage.

In order to address the dilemma, we propose an on-policy skill-based RL method named dynamical skill refinement (DSR) which dynamically refines the skills under the optimization objective unified with the high-level policy. We theoretically prove that this objective guarantees the performance improvement in TA-MDP. More importantly, we innovatively consider skill-based RL as the proxy of the original RL task, and theoretically prove that optimizing the performance in TA-MDP is equivalent to optimizing a lower bound of the performance in the original MDP. We simultaneously update high-level policy and skills in an on-policy RL manner, so as to circumvent the temporal abstraction shift. Furthermore, we point out that directly refining skills may lead to skill space collapse which brings in performance collapse. This is because all skills are embedded into the latent space of the same parametric low-level policy. Changing the behavior of one skill may cause the behaviors of other skills to be changed as well. Therefore, we augment our method with the dynamical skill refinement mechanism which ensures that the skills maintain their original behaviors before being explored. We empirically validate the effectiveness of our method and its advantages over the state-of-the-art (SOTA) methods in multiple sparse-reward tasks of the robotics manipulation domain, and prove the necessity of dynamical skill refinement.

We summarize the contributions of this paper as follows: (1) we propose a unified optimization objective for both high-level policy and skills, and theoretically analyze its effectiveness, (2) we devise an on-policy skill-based RL method and a dynamical skill refinement mechanism which avoids skill space collapse, and (3) we empirically validate the effectiveness and superiority of our method in multiple sparse-reward tasks.

## 2 RELATED WORK

### 2.1 HIERARCHICAL REINFORCEMENT LEARNING

In hierarchical reinforcement learning (HRL), the policy stitches temporally extended behaviors rather than primitive actions to be the behavior of solving tasks Pateria et al. (2021); Li et al. (2019; 2022); Kipf et al. (2019); Zhang & Whiteson (2019); Yang et al. (2023). The low-level policy is usually conditioned on the latent space which serves as the action space of the high-level policy. Both the high-level and the low-level policies can be learned from the experience obtained from the agent's interacting with the environment Bacon et al. (2017); Haarnoja et al. (2018a); Levy et al. (2019). In practical applications, it is a mainstream approach to extract the low-level policy from the demonstration dataset and then learn to recompose the behavior modes of the low-level policy to solve the task Shankar et al. (2019); Krishnan et al. (2017); Kipf et al. (2019); Nachum et al. (2018).

### 2.2 SKILL-BASED REINFORCEMENT LEARNING

Skill-based reinforcement learning (skill-based RL) extracts reusable behavior modes from the demonstration dataset which are used to solve the downstream task Hausman et al. (2018); Pertsch et al. (2021; 2022). These behavior modes are considered as skills and embedded into the latent space, which is typically through a variational auto-encoder (VAE) Kingma & Welling (2013). During online learning, the high-level policy learns to select the optimal skills to solve the downstream

task. The high-level policy learns to choose temporally extended skills rather than single-timestep actions, which shortens the task horizon and reduces the exploration cost. Prior works generally focus on extracting the skill prior from the demonstrations to improve the exploration efficiency Walke et al. (2023); Xu et al. (2022); Nam et al. (2022); Pertsch et al. (2021; 2022). These works assume that the skills can be approximately considered as optimal for the downstream task. However, this assumption can hardly be met when the downstream task is different from the tasks used to generate the demonstration dataset. It is intuitive and natural to refine the skills while learning the high-level policy Rana et al. (2023); Hao et al. (2024), but this issue is still not well addressed.

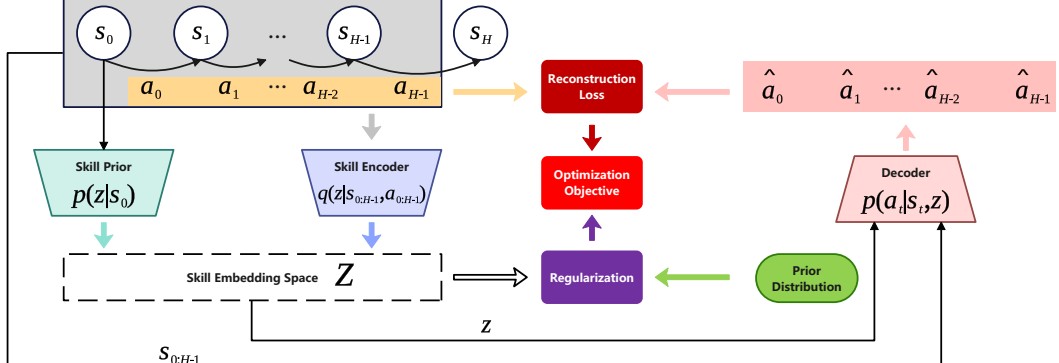

Figure 1: State-action segments of fixed length are sampled from the demonstration dataset. These segments are embedded into the latent space by the auto-encoder. The decoder reconstructs the action segment based on the state segment and the latent variable. We not only reduce the reconstruction loss, but also make the distribution of skill embedding close to the prior distribution. Skill prior learns to predict the skill embedding of the state-action segment based on only the first state.

# 3 PRELIMINARY

## 3.1 TA-MDP

In skill-based RL, the high-level policy $\pi^h$ and the low-level policy $\pi^l$ whose latent space embeds the skills $\{z\}$ constitute the entire hierarchical policy. The high-level policy selects the skills from the latent space $Z$ which are the behavior modes of the low-level policy $\pi^l$ conditioned on the latent variables. Each skill manifests as a sequence of primitive actions $\{a_t, a_{t+1}, ..., a_{t+H-1}\}$ over a fixed horizon $H$. Once a skill $z_t$ is selected from $\pi^h(\cdot|s_t)$ by the high-level policy, the action distributions at the next $H$ consecutive timesteps will be $a_{t+i} \sim \pi^l(\cdot|s_{t+i}, z_t), i \in [0, H-1]$.

In reinforcement learning, the task is formulated in a Markov decision process (MDP) $(\mathcal{S}, \mathcal{A}, p, r, \gamma)$ which consists of the state space $\mathcal{S}$, the action space $\mathcal{A}$, the transition dynamics $p(s'|s, a)$, the reward function $r(s, a)$ and the discount factor $\gamma$. However, in skill-based RL, each skill $z$ selected by the high-level policy $\pi^h$ is over the fixed horizon $H$, so $\pi^h$ is actually learning in a TA-MDP $(\mathcal{S}, \mathcal{A}, p_{\pi^l, H}, \tilde{r}, \tilde{\gamma})$. The transition dynamics of TA-MDP $p_{\pi^l, H}(s'|s, z)$ is the distribution over the state $s_{t+H}$ after $H$ timesteps conditioned on the state $s_t$ and the selected skill $z_t$, $p(s_{t+H} = s'|s_t = s, z_t = z, \pi^l)$. The reward function of TA-MDP $\tilde{r}(s_t, z_t) = \sum_{i=0}^{H-1} r_{t+i}$ is the sum of $H$ consecutive rewards obtained by executing skill $z_t$ from state $s_t$.

## 3.2 EXTRACT SKILLS THROUGH VAE

In order to embed the skills into the latent space of low-level policy, it is common to sample state-action segments from the demonstration dataset and encode them as latent variables through the variational auto-encoder (VAE) Pertsch et al. (2021; 2022); Rana et al. (2023). The decoder conditioned on state and latent variable will reconstruct action segments. The sum of the loss of reconstructed action segment and the KL-divergence between the latent variable distribution and a prior distribution is the complete optimization objective. The trained decoder will serve as the low-level

policy. Usually, a skill prior that is conditioned on the first state and attempts to predict the same latent variable will also be learned to improve the exploration efficiency in the online learning stage. The skill prior can be used as the initial high-level policy and be used for biasing the exploration to the regions of potentially valuable skills. This extraction process is sketched in Figure 1.

# 4 UPDATE HIGH-LEVEL POLICY AND SKILLS UNDER UNIFIED OPTIMIZATION OBJECTIVE

We take the expected sum of the discounted rewards in the TA-MDP as the unified optimization objective of both learning high-level policy and refining skills. In particular, the optimization objective of skills is the sum of inner-skill single-timestep rewards of MDP and the subsequent discounted $H$-timestep rewards of TA-MDP. We prove that this optimization objective can ensure the performance improvement of the entire hierarchical policy in TA-MDP. Furthermore, we innovatively consider skill-based RL as a proxy task of RL, and prove that optimizing the expected sum of discounted $H$-timestep rewards in the TA-MDP is equivalent to optimizing a lower bound of the expected sum of the discounted single-timestep rewards in the original MDP, which provides the effectiveness.

## 4.1 UNIFIED OPTIMIZATION OBJECTIVE AND UPDATE FORMULAE

In skill-based RL, the task of high-level policy is to maximize the expected sum of discounted $H$-timestep rewards $\tilde{r}(s, z)$ in TA-MDP, which can be formulated as follows:

$$\max_{\pi^h} \mathbb{E}_{\pi^h, \pi^l} \Big[ \sum_{\hat{t}} \tilde{\gamma}^{\hat{t}} \cdot \tilde{r}_{\hat{t}} \Big], \text{ where } \tilde{r}_{\hat{t}} = \sum_{t=\hat{t} \cdot H}^{\hat{t} \cdot H + H - 1} r_t. \tag{1}$$

$\hat{t}$ and $t$ are the timestep index variables of TA-MDP and MDP respectively. The $H$-timestep reward $\tilde{r}$ in the TA-MDP is the sum of $H$ consecutive single-timestep rewards $r$ in the original MDP. Unlike the high-level policy, skills predict a primitive action at each timestep. We design the optimization objective of skills to be the sum of all the inner-skill single-timestep rewards and the discounted $H$-timestep rewards after the skill. We formulate this objective as follows:

$$\max_{\pi^l} \mathbb{E}_{\pi^h, \pi^l} \Big[ \sum_{i=\hat{t} \cdot H}^{\hat{t} \cdot H + H - 1} r_i + \sum_{j=\hat{t}+1}^{\infty} \tilde{\gamma}^{j-\hat{t}} \cdot \tilde{r}_j \Big]. \tag{2}$$

To formulate the optimization objective for policy iteration, we define the state value function $V_{\pi^h, \pi^l}^h(s)$ and state-skill value function $Q_{\pi^h, \pi^l}^h(s, z)$ in the TA-MDP which are analogous to the state value function and state-action value function in the original MDP. Therefore, the policy-iteration optimization objective of the high-level policy regarding a state $s$ can be formulated as follows:

$$\max_{\pi^h} \mathbb{E}_{z \sim \pi^h(\cdot|s)}[Q_{\pi^h, \pi^l}^h(s, z)]. \tag{3}$$

The policy-iteration optimization objective of the skill $z$ executed from state $s$ at timestep $\hat{t} \cdot H$ can be formulated as follows:

$$\max_{\pi^l} \Big[ \mathbb{E}_{a_t \sim \pi^l(\cdot|s_t, z)} \Big[ \sum_{t=\hat{t} \cdot H}^{\hat{t} \cdot H + H - 1} r_t \Big] + \tilde{\gamma} \cdot \mathbb{E}_{s' \sim p_{\pi^l, H}(\cdot|s, z)}[V_{\pi^h, \pi^l}^h(s')] \Big], \tag{4}$$

where $p_{\pi^l, H}(\cdot|s, z)$ is the distribution of reached state after executing $z$ from $s$ for $H$ timesteps.

We parameterize $\pi^h(z|s)$ and $\pi^l(a|s, z)$ as $\pi_\phi^h(z|s)$ and $\pi_\theta^l(a|s, z)$. Then, the recursive update formulae for $\pi_\phi^h(z|s)$ and $\pi_\theta^l(a|s, z)$ are as follows:

$$\forall s \in \mathcal{S}, \pi_{\phi'}^h(\cdot|s) \leftarrow \arg \max_{\pi^h(\cdot|s)} \mathbb{E}_{z \sim \pi^h(\cdot|s)}[Q_{\pi_\phi^h, \pi_\theta^l}^h(s, z)], \tag{5}$$

$$\forall s_t \in \mathcal{S}, \forall z \in Z, \pi_{\theta'}^l(\cdot|s_t, z) \leftarrow \arg \max_{\pi^l(\cdot|s_t, z)} \Big[ \mathbb{E}_{a_t \sim \pi^l(\cdot|s_t, z)} \Big[ \sum_{t=\hat{t} \cdot H}^{\hat{t} \cdot H + H - 1} r_t \Big] + \tag{6}$$

$$\tilde{\gamma} \cdot \mathbb{E}_{s' \sim p_{\pi^l, H}(\cdot|s_{\hat{t} \cdot H}, z)}[V_{\pi_\phi^h, \pi_\theta^l}^h(s')] \Big],$$

where $\phi'$ and $\theta'$ are the updated version of parameters $\phi$ and $\theta$.

## 4.2 PERFORMANCE IMPROVEMENT AND CONVERGENCE IN TA-MDP

We prove that the updates of $\pi_\phi^h, \pi_\theta^l$ which follow Equations 5 and 6 guarantee the performance improvement in TA-MDP. Under these updates, the performance in TA-MDP will converge.

To prove that there is a monotonic increase in the state value function, we first illustrate Lemma 1:

**Lemma 1.** $\forall s \in \mathcal{S}$, we denote the value of first predicting a skill $z$ with $\pi_{\phi'}^h$, executing $\pi_{\theta'}^l(\cdot|s, z)$ for $H$ timesteps and then executing $\pi_\phi^h, \pi_\theta^l$ by $\tilde{V}_{\pi_{\phi'}^h, \pi_{\theta'}^l, \pi_\phi^h, \pi_\theta^l}^h(s)$. The following inequality relations are valid, if we follow the update formulae in Equations 5 and 6:

$$\forall s \in \mathcal{S}, \tilde{V}_{\pi_{\phi'}^h, \pi_{\theta'}^l, \pi_\phi^h, \pi_\theta^l}^h(s) \geq V_{\pi_\phi^h, \pi_\theta^l}^h(s). \tag{7}$$

*See Appendix A.1 for proof.*

Based on Lemma 1, we prove the monotonic performance increase in the TA-MDP:

**Theorem 1.** *If we update $\pi_\phi^h$ and $\pi_\theta^l$ following Equations 5 and 6, then:*

$$\forall s \in \mathcal{S}, V_{\pi_{\phi'}^h, \pi_{\theta'}^l}^h(s) \geq V_{\pi_\phi^h, \pi_\theta^l}^h(s). \tag{8}$$

*See Appendix A.2 for proof.*

Due to the performance improvement in Theorem 1, we get the convergence of performance:

**Theorem 2.** *With the policy update formulae in Equations 5 and 6, the state value function $V_{\pi_{\phi_k}^h, \pi_{\theta_k}^l}^h(s)$ will finally converge. $\pi_{\phi_k}^h, \pi_{\theta_k}^l$ are the $k$-th version of high-level policy and low-level policy respectively. See Appendix A.3 for proof.*

## 4.3 EFFECTIVENESS OF LEARNING IN TA-MDP

To analyze the effectiveness of our optimization objective, we propose that the essential purpose of skill-based RL is to make the behavior of the entire hierarchical policy approximate to that of the optimal policy in the original MDP. Unlike prior works Pertsch et al. (2021; 2022); Rana et al. (2023); Hao et al. (2024) which considers skill-based RL as a paradigm parallel to RL, our perspective establishes a connection between skill-based RL and RL. We sketch this perspective in Figure 2 which motivates us to analyze the relationship between the performance in the TA-MDP and that in the original MDP. We prove that optimizing the performance of a hierarchical policy in the TA-MDP is equivalent to optimizing a lower bound of its performance in the original MDP.

Given the original MDP $(\mathcal{S}, \mathcal{A}, p, r, \gamma)$ and hierarchical policy $(\pi^h, \pi^l)$, we denote the corresponding TA-MDP by $(\mathcal{S}, \mathcal{A}, p_{\pi^l, H}, \tilde{r}, \tilde{\gamma})$. The performance of $(\pi^h, \pi^l)$ in the original MDP and that in the TA-MDP can be formulated into analogous expanded forms:

$$V_{\pi^h, \pi^l}(s) = \int \sum_{\triangle t=0}^\infty \rho_{\triangle t}(s', a'|s, \pi^h, \pi^l) \gamma^{\triangle t} \cdot r(s', a') da' ds', \tag{9}$$

$$V_{\pi^h, \pi^l}^h(s) = \int \sum_{\triangle t=0}^\infty \rho_{\triangle t}(s', a'|s, \pi^h, \pi^l) \tilde{\gamma}^{\lfloor \triangle t/H \rfloor} \cdot r(s', a') da' ds', \tag{10}$$

$\rho_{\triangle t}(s', a'|s, \pi^h, \pi^l)$ is the state-action distribution after executing $\pi^h, \pi^l$ for $\triangle t$ timesteps from $s$.

Since the discount factor is manually specified, we can let $\tilde{\gamma} = \gamma^H$. We illustrate in Theorem 3 that optimizing the performance of the hierarchical policy in the TA-MDP is equivalent to optimizing its performance lower bound in the MDP if the reward function gives a positive feedback only when the task is completed.

**Theorem 3.** *If the MDP $(\mathcal{S}, \mathcal{A}, p, r, \gamma)$ and the TA-MDP $(\mathcal{S}, \mathcal{A}, p_{\pi^l, H}, \tilde{r}, \tilde{\gamma})$ satisfy that $\tilde{\gamma} = \gamma^H$, then $\forall s \in \mathcal{S}, \tilde{\gamma} \cdot V_{\pi^h, \pi^l}^h(s) \leq V_{\pi^h, \pi^l}(s)$. It means that optimizing $V_{\pi^h, \pi^l}^h(s)$ is equivalent to optimizing a lower bound of $V_{\pi^h, \pi^l}(s)$. See Appendix A.4 for proof.*

Since we have proved in Theorem 1 that our optimization objective can guarantee the monotonic increase in $V_{\pi^h, \pi^l}^h(s)$, refining skills under this objective is equivalent to optimizing the lower bound of $V_{\pi^h, \pi^l}(s)$, which illustrates the effectiveness of our method.

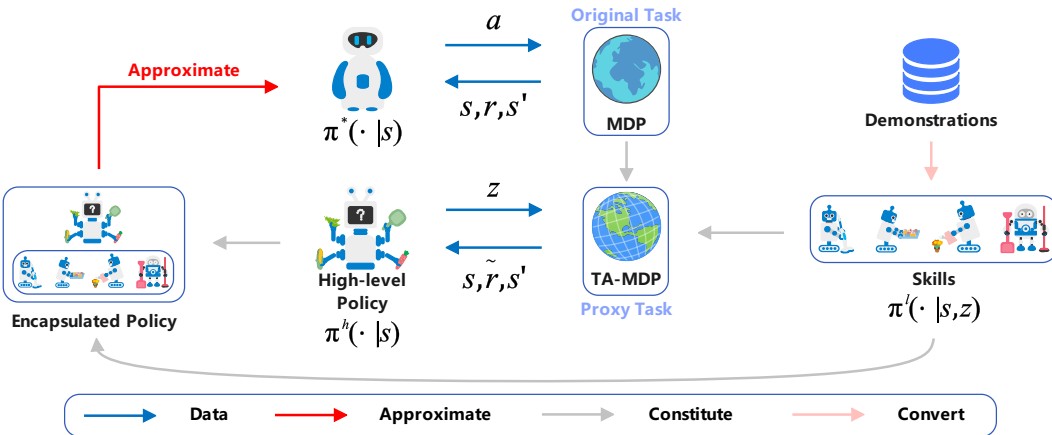

Figure 2: In skill-based RL, the extracted skills and the original MDP constitute the TA-MDP. We regard learning the hierarchical policy in TA-MDP as a proxy task of learning a flat policy in the original MDP. We consider the hierarchical policy as a whole, which gives an action at each timestep, just like the flat policy.

# 5 UPDATE HIERARCHICAL POLICY WITH DYNAMICAL SKILL REFINEMENT IN AN ON-POLICY RL MANNER

We first illustrate how to optimize our objective in an on-policy RL manner. Then, we elaborate on the dynamical skill refinement mechanism and explain its necessity.

## 5.1 OPTIMIZING THE OBJECTIVE IN AN ON-POLICY RL MANNER

Since high-level policy is learned in TA-MDP, we can directly apply the on-policy RL algorithms Schulman et al. (2015; 2017) to learn it. Therefore, we focus on illustrating how to refine the skills.

We first observe that the policy-iteration optimization objective of skills in Equation 6 can be decomposed into two components: (1) the expected sum of the inner-skill rewards given by original MDP and (2) the expected sum of the discounted $H$-timestep rewards given by TA-MDP. In addition, a trajectory is composed of transition segments with the length of $H$ timesteps generated by multiple skills. Therefore, we innovatively consider the $H$-timestep transition segment generated by a skill in the trajectory as a rollout and equivalently add the sum of subsequent discounted $H$-timestep rewards to the last reward of the rollout. This idea is sketched in Figure 3. We can directly apply the on-policy RL algorithms on the resulted rollout to refine the corresponding skill.

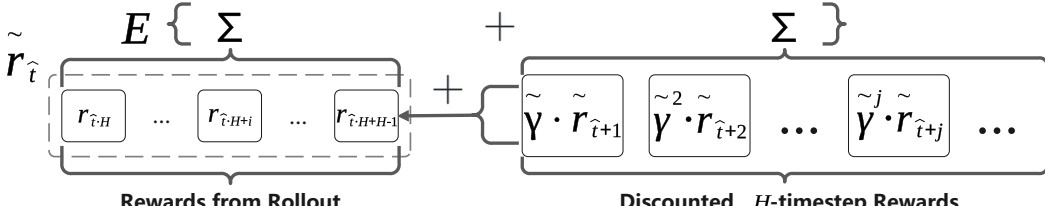

Figure 3: The $H$ consecutive transitions generated by the same skill can be seen as a rollout. We add the sum of discounted $H$-timestep rewards $\tilde{r}$ to the last reward $r_{\hat{t} \cdot H + H - 1}$ in it. Then, we apply on-policy RL methods on the resulted rollout to optimize the skill. In this way, we equivalently optimize the expected sum of inner-skill rewards and discounted $H$-timestep rewards after the skill.

## 5.2 DYNAMICAL SKILL REFINEMENT MECHANISM

We first explain why directly refining skills causes skill space collapse which manifests as performance collapse. We then elaborate on the dynamical skill refinement mechanism.

As we have mentioned in Section 3, all the skills are usually embedded into the latent space of the same parametric low-level policy. If we refine a skill, the behaviors of other skills may change stochastically. For a state, the optimal and near-optimal skills usually occupy only a small region of the entire skill space, which means that these skills may have experienced lots of stochastic updates and lost their original behaviors before being sampled by the high-level policy. As shown in Figure 4, good candidate skills may no longer exist in a specific state. We name this phenomenon skill space collapse. See Appendix B for the visualization of skill space collapse in specific task.

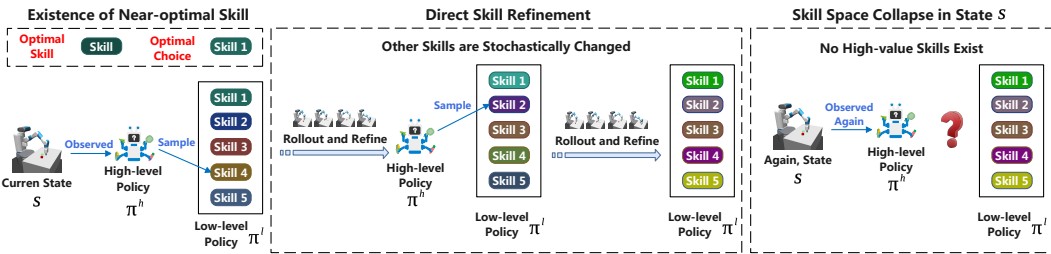

Figure 4: Before predicting the optimal skill, the high-level policy usually needs to sufficiently explore the skill space in a given state. If we refine selected skills directly, a terrible selected skill will not quickly approach the optimal skill, but other skills will change stochastically. The best skill in the skill library for a state may be destroyed after consecutive rollouts and skill refinements. When encountering the state again, there are actually no good candidate skills that can be explored.

Intuitively, before a skill is executed in a specific state, its behavior in the state should remain the same as when it was extracted. In addition, since we have theoretically proved the effectiveness of skill refinement, we believe that sufficiently refining a skill in a state can overcome the potential damage to its behavior caused by stochastic changes. We learn skill refinements into a separate residual policy Rana et al. (2023) rather than the low-level policy, which can preserve the extracted behaviors of the skills. The action increment predicted by the residual policy is added to the action predicted by the low-level policy to form the practical action given by the skill. We assign a dynamical weight to action increment, that is, when a skill has been sufficiently refined in a state, the action increment for the state-skill will be given a high weight, otherwise, it should be given a low weight. We name this measure dynamical skill refinement (DSR) mechanism.

We use random network distillation (RND) Burda et al. (2019) to measure whether the behavior of a skill in a state has been sufficiently refined. It involves two randomly initialized neural networks, namely, fixed target network and variable predictor network. The target network takes a state-skill to an embedding $f : \mathcal{S} \times \mathcal{Z} \to \mathbb{R}^k$. The predictor network $\hat{f} : \mathcal{S} \times \mathcal{Z} \to \mathbb{R}^k$ is trained to minimize the MSE loss $||\hat{f}((s,z);\xi) - f((s,z))||^2$ with respect to its parameters $\xi_{\hat{f}}$. Every time we refine the behavior of skill $z$ in state $s$, we optimize the prediction error of the predictor network on the state-skill $(s, z)$. We expect that the prediction errors on those state-skills on which the predictor network has been trained for many times are obviously lower than the prediction errors on the novel state-skills. We can map the prediction error on a state-skill to the weight of action increment. See Appendix C for the complete algorithm incorporating the dynamical skill refinement.

## 6 EXPERIMENTS

In the experiments, we first compare the performance of DSR to the SOTA methods in solving sparse-reward tasks. Then, we conduct the ablation analysis on the dynamical skill refinement mechanism and the involved hyper-parameters.

We adopt the 4 robotic manipulation tasks proposed in ReSkill Rana et al. (2023). These tasks involve a manipulator arm that can fetch objects, push objects and even move objects using tools.

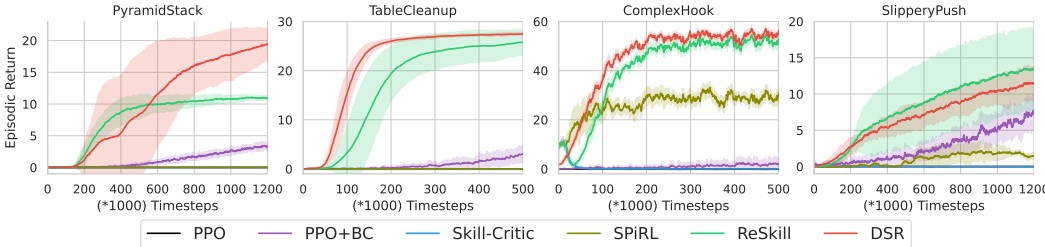

Figure 5: These tasks are shown in the upper part. (1) **SlipperyPush**: push the cube to the target position. (2) **TableCleanup**: Grab the cube and place it on the tray. (3) **PyramidStack**: Grab the cube and place it on the larger cube. (4) **ComplexHook**: Use the hook to move the object to the target position. The lower part shows the skill sequence to complete ComplexHook.

We show these tasks and how the arm solves a task through combining multiple skills in Figure 5. The demonstration datasets are generated by hand-scripted controller's interacting with the environments. The physically modified versions of these tasks serve as the downstream tasks. These physical modifications necessitate skill refinements. See Appendix D for the details of experiments.

## 6.1 COMPARISON WITH SOTA METHODS

Figure 6: DSR achieves the highest performance, except for slightly lower performance than ReSkill in SlipperyPush, but DSR shows significantly lower variance. Due to the sub-optimality of skills for downstream tasks, SPiRL can only solve ComplexHook and SlipperyPush, and its performance is limited. Without the warm start phase based on SPiRL, Skill-Critic can only solve ComplexHook in the initial phase and eventually suffers performance collapses in all the tasks. PPO+BC suffers from low sample efficiency, and PPO doesn't even improve performance at all.

We compare our method with several SOTA skill-based RL methods. **SPiRL** Pertsch et al. (2021) extracts temporally extended behaviors along with a skill prior from the demonstration dataset and assume these skills are approximately optimal for the downstream task. **Skill-Critic** Hao et al. (2024) ignores the temporal abstraction shift and updates the entire hierarchical policy in an off-policy RL manner. **ReSkill** Rana et al. (2023) learns skill refinement into the residual policy and updates the entire hierarchical policy in an on-policy RL manner, which circumvents the temporal abstraction shift. We also compare our method with **PPO** Schulman et al. (2017) and **PPO+BC**. The latter one fine-tunes a policy initialized by behavior cloning through PPO.

For fair comparison, our method (DSR), SPiRL, ReSkill, and Skill-Critic share the same model architecture of VAE. The warm start phase of Skill-Critic with the help of SPiRL is removed. The trick of ReSkill that gradually gives higher weight to the prediction of residual policy with the number of updates is retained. The results are averaged over four random seeds.

The performance curves are presented in Figure 6. SPiRL can only solve ComplexHook and SlipperyPush with low performance illustrating that the extracted skills are sub-optimal for downstream tasks with modifications on the physical properties. Only in the initial stage of ComplexHook, the performance of Skill-Critic can be temporarily improved, and the performance collapses in all other

times, which shows that the temporal abstraction shift can not be ignored. ReSkill learns the skill refinements into the residual policy, which preserves the original behaviors of the extracted skills to some extent. However, it still suffers significant performance degradation in ComplexHook, which indicates that skill space is still compromised. In contrast, DSR can steadily improve the performance throughout the online learning stage, which verifies the effectiveness of our optimization objective and indicates that skill space collapse is completely avoided. Both PPO and PPO+BC suffer from extremely slow performance improvement. On the contrary, DSR can improve the performance to a high level at a fast speed, which illustrates the necessity of temporal abstraction.

## 6.2 ABLATION ANALYSIS OF DYNAMICAL SKILL REFINEMENT

In the ablation analysis, we analyze the necessity of dynamical skill refinement, the improvement of skills' optimality, and whether DSR is sensitive to hyper-parameters.

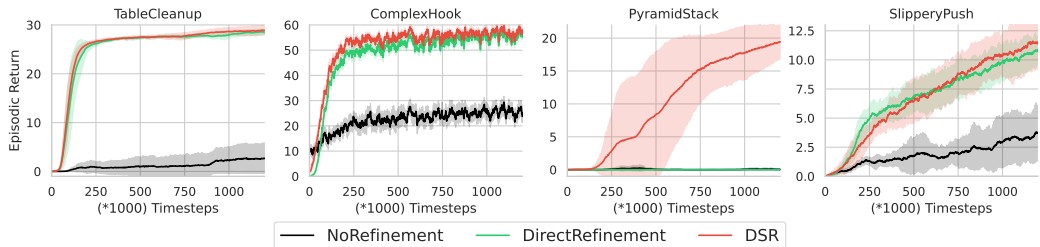

Figure 7: If skills are kept fixed, the asymptotic performance are limited. The extracted skills are not well initialized for PyramidStack, so the skill space collapse caused by direct skill refinement destroys the low-level policy and the performance is not improved at all. In contrast, when we fine-tune the entire hierarchical policy with dynamical skill refinement mechanism, the performance is improved significantly than fixing the skills. Furthermore, our method can still improve the performance in PyramidStack where direct skill refinement fails, which indicates that the dynamical skill refinement mechanism prevents low-level policy from being destroyed by skill space collapse.

To illustrate the need for skill refinement and the need for the dynamical skill refinement mechanism, we compare the performance of our complete method with the performance of two other cases. In the first case, the skills are not refined (**NoRefinement**). In the second case, the dynamical skill refinement mechanism is removed and the skills are refined directly (**DirectRefinement**). These performance curves are presented in Figure 7. We find that if the skills are not refined, the asymptotic performance is limited even if the skills are well initialized, which is shown in TableCleanup, ComplexHook and SlipperyPush. If the extracted skills are not well initialized for the downstream task, then the entire hierarchical policy completely fails to solve the task, which is shown in PyramidStack. In contrast, our method achieves obviously higher performance in all the tasks, which illustrates the necessity of refining skills. When the extracted skills are well initialized for the downstream task, the potential skill space collapse may not be enough to destroy the low-level policy, and refining the skills directly can still improve the performance. However, if the extracted skills are not well initialized for the downstream task, the skill space collapse will destroy the low-level policy and the performance can not be improved at all, which is shown in PyramidStack. In contrast, if skills are not well initialized, the dynamical skill refinement mechanism can effectively prevent the fragile low-level policy from being destroyed by skill space collapse. Moreover, in contrast to successful cases of direct skill refinement, we find that though the dynamical skill refinement is a conservative measure, it does not reduce the sample efficiency.

To analyze the improvement in the optimality of skills, we compare the performance achieved by the same sufficiently trained high-level policy when the skills are refined and when they are not. The performance brought about by the skills with refinement (**With**) and the performance brought about by the skills without refinement (**Without**) are shown in Figure 8. Since the high-level policy is the same, the performance improvement reflects the improvement of skills' optimality.

Finally, we empirically demonstrate the insensitivity of our method to hyper-parameters. The mapping to the weights of action increment from prediction errors ($S(x) = \alpha \frac{1}{1+e^{-k(x-c)}}$, see Appendix D.3 for details, $\alpha$ is simply set to 1) is mainly determined by two parameters, $c$ and $k$ which are

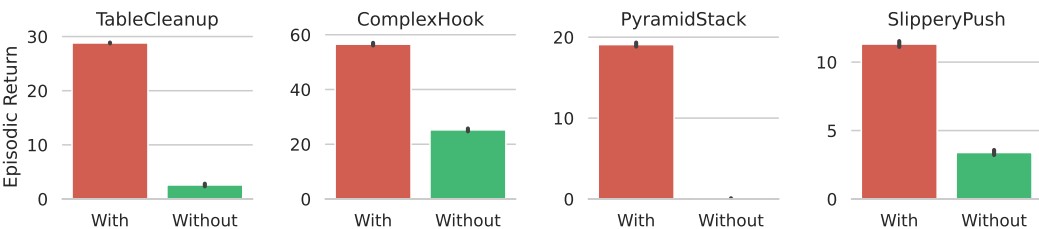

Figure 8: Under the same high-level policy, skills with refinement lead to higher performance, which obviously benefits from the improved optimality of skills.

hand-scripted. We determined the values of $k$ and $c$ as $-300$ and $0.025$ for TableCleanup according to the method in the Appendix D.4. Similarly, the total rounds of variable predictor network's being trained in every epoch is also a hyper-parameter. We set the number of rounds to $10$ which is an empirical value. To demonstrate that dynamical skill refinement is insensitive to these hyper-parameters, we respectively adjust them within a certain range and show in Figure 9 the performance curves of these cases in TableCleanup. It is evident that adjusting these hyper-parameters within the certain intervals centered in the manually specified values results no significant performance decline, indicating DSR's insensitivity to hyper-parameters.

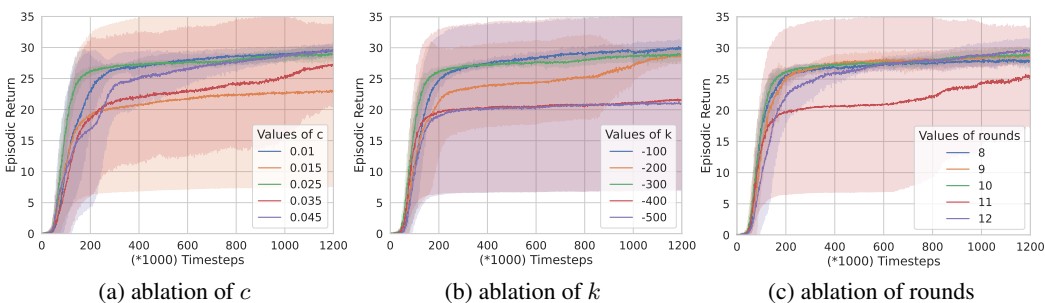

(a) ablation of $c$          (b) ablation of $k$          (c) ablation of rounds

Figure 9: We conduct ablation analysis on the three hyper-parameters separately in TableCleanup. Modifying the hyper-parameters in the certain intervals centered in the hand-scripted values results in no significant changes in performance.

## 7 CONCLUSION

In this paper, we study the skill-based RL method of updating both high-level policy and skills. We innovatively consider skill-based RL as the proxy task of RL, and theoretically prove that optimizing the performance of the hierarchical policy in TA-MDP is equivalent to optimizing the performance lower bound in original MDP. We devise the optimization objective of skills which is unified with high-level policy and propose to update the entire hierarchical policy in an on-policy RL manner so as to circumvent the temporal abstraction shift. We propose for the first time that direct skill refinement leads to the skill space collapse phenomenon since all the skills are embedded into the latent space of the same parametric policy. To address skill space collapse, we propose the dynamical skill refinement mechanism which can be simply integrated into our algorithm. Dynamical skill refinement guarantees that the behavior of low-level policy on a given state-skill remains approximate to the extracted behavior before it is sufficiently refined, thus avoiding skill space collapse.

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

## A DETAILED THEORETICAL PROOF

### A.1 PROOF OF LEMMA 1

**Lemma 1.** $\forall s \in \mathcal{S}$, we denote the value of first predicting a skill $z$ with $\pi_{\phi'}^h$, executing $\pi_{\theta'}^l(\cdot|s,z)$ for $H$ timesteps and then executing $\pi_\phi^h, \pi_\theta^l$ by $\tilde{V}_{\pi_{\phi'}^h, \pi_{\theta'}^l, \pi_\phi^h, \pi_\theta^l}^h(s)$. The following inequality relations are valid, if we follow the update formulae in Equations 5 and 6:

$$\forall s \in \mathcal{S}, \tilde{V}_{\pi_{\phi'}^h, \pi_{\theta'}^l, \pi_\phi^h, \pi_\theta^l}^h(s) \geq V_{\pi_\phi^h, \pi_\theta^l}^h(s). \tag{11}$$

*Proof.*

$$\forall s \in \mathcal{S}, \tilde{V}_{\pi_{\phi'}^h, \pi_{\theta'}^l, \pi_\phi^h, \pi_\theta^l}^h(s) = \tag{12}$$

$$\mathbb{E}_{z \sim \pi_{\phi'}^h(\cdot|s)}[\mathbb{E}_{a_i \sim \pi_{\theta'}^l(\cdot|s_i,z), s_0=s}[\sum_{i=0}^{H-1} r_i] + \tilde{\gamma} \cdot \mathbb{E}_{s' \sim p_{\pi_{\theta'}^l, H}(\cdot|s,z)}[V_{\pi_\phi^h, \pi_\theta^l}^h(s')]].$$

According to the update formula of skills in Equation 6, we can infer that it is larger than the following equation:

$$\geq \mathbb{E}_{z \sim \pi_{\phi'}^h(\cdot|s)}[\mathbb{E}_{a_i \sim \pi_\theta^l(\cdot|s_i,z), s_0=s}[\sum_{i=0}^{H-1} r_i] + \tilde{\gamma} \cdot \mathbb{E}_{s' \sim p_{\pi_\theta^l, H}(\cdot|s,z)}[V_{\pi_\phi^h, \pi_\theta^l}^h(s')]] \tag{13}$$

$$= \mathbb{E}_{z \sim \pi_{\phi'}^h(\cdot|s)}[Q_{\pi_\phi^h, \pi_\theta^l}^h(s,z)]. \tag{14}$$

By applying the properties of the high-level policy update formula in Equation 5, we can further infer that:

$$\geq \mathbb{E}_{z \sim \pi_\phi^h(\cdot|s)}[Q_{\pi_\phi^h, \pi_\theta^l}^h(s,z)]. \tag{15}$$

Obviously, it is the value of constantly executing the $\pi_\phi^h, \pi_\theta^l$ from state $s$, namely $V_{\pi_\phi^h, \pi_\theta^l}^h(s)$. In summary, we get the following inequality relationship:

$$\forall s \in \mathcal{S}, \tilde{V}_{\pi_{\phi'}^h, \pi_{\theta'}^l, \pi_\phi^h, \pi_\theta^l}^h(s) \geq V_{\pi_\phi^h, \pi_\theta^l}^h(s). \tag{16}$$

$\square$

### A.2 PROOF OF THEOREM 1

**Theorem 1.** If we update $\pi_\phi^h$ and $\pi_\theta^l$ following Equations 5 and 6 respectively, then $\forall s \in \mathcal{S}, V_{\pi_{\phi'}^h, \pi_{\theta'}^l}^h(s) \geq V_{\pi_\phi^h, \pi_\theta^l}^h(s)$.

*Proof.* We first rewrite $\tilde{V}_{\pi_{\phi'}^h, \pi_{\theta'}^l, \pi_\phi^h, \pi_\theta^l}^h(s)$ in its expansion:

$$\tilde{V}_{\pi_{\phi'}^h, \pi_{\theta'}^l, \pi_\phi^h, \pi_\theta^l}^h(s) = \mathbb{E}_{z \sim \pi_{\phi'}^h(\cdot|s)}[\mathbb{E}_{a_i \sim \pi_{\theta'}^l(\cdot|s_i,z), s_0=s}[\sum_{i=0}^{H-1} r_i] + \tilde{\gamma} \cdot \mathbb{E}_{s' \sim p_{\pi_{\theta'}^l, H}(\cdot|s,z)}[V_{\pi_\phi^h, \pi_\theta^l}^h(s')]].$$

$$\tag{17}$$

We can enlarge $V_{\pi_\phi^h, \pi_\theta^l}^h(s')$ to $\tilde{V}_{\pi_{\phi'}^h, \pi_{\theta'}^l, \pi_\phi^h, \pi_\theta^l}^h(s')$ according to Lemma 1:

$$\leq \mathbb{E}_{z \sim \pi_{\phi'}^h(\cdot|s)}[\mathbb{E}_{a_i \sim \pi_{\theta'}^l(\cdot|s_i,z), s_0=s}[\sum_{i=0}^{H-1} r_i] + \tilde{\gamma} \cdot \mathbb{E}_{s' \sim p_{\pi_{\theta'}^l, H}(\cdot|s,z)}[\tilde{V}_{\pi_{\phi'}^h, \pi_{\theta'}^l, \pi_\phi^h, \pi_\theta^l}^h(s')]]. \tag{18}$$

We can expand the $\tilde{V}^h_{\pi^h_{\phi'},\pi^l_{\theta'},\pi^h_\phi,\pi^l_\theta}(s')$ and get:

$$= \mathbb{E}_{z\sim\pi^h_{\phi'}(\cdot|s)}[\mathbb{E}_{a_i\sim\pi^l_{\theta'}(\cdot|s_i,z),s_0=s}[\sum_{i=0}^{H-1} r_i]+ \tag{19}$$

$$\tilde{\gamma}\cdot\mathbb{E}_{s'\sim p_{\pi^l_{\theta'},H}(\cdot|s,z)}[\mathbb{E}_{z'\sim\pi^h_{\phi'}(\cdot|s')}[\mathbb{E}_{a_j\sim\pi^l_{\theta'}(\cdot|s_j,z'),s_H=s'}[\sum_{j=H}^{2H-1} r_j] + \tilde{\gamma}\cdot\mathbb{E}_{s''\sim p_{\pi^l_{\theta'},H}(\cdot|s',z')}[V^h_{\pi^h_\phi,\pi^l_\theta}(s'')]]]].$$

If we expand $\tilde{V}^h_{\pi^h_{\phi'},\pi^l_{\theta'},\pi^h_\phi,\pi^l_\theta}(\cdot)$ and enlarge $V^h_{\pi^h_\phi,\pi^l_\theta}(\cdot)$ to $\tilde{V}^h_{\pi^h_{\phi'},\pi^l_{\theta'},\pi^h_\phi,\pi^l_\theta}(\cdot)$ infinitely and repeatedly, we will obviously get:

$$\leq ... \leq ... = V^h_{\pi^h_{\phi'},\pi^l_{\theta'}}(s). \tag{20}$$

Taking Lemma 1 into account, we get the following inequality:

$$\forall s \in \mathcal{S}, V^h_{\pi^h_\phi,\pi^l_\theta}(s) \leq \tilde{V}^h_{\pi^h_{\phi'},\pi^l_{\theta'},\pi^h_\phi,\pi^l_\theta}(s) \leq V^h_{\pi^h_{\phi'},\pi^l_{\theta'}}(s). \tag{21}$$

In summary, the following inequality relationship is valid:

$$\forall s \in \mathcal{S}, V^h_{\pi^h_{\phi'},\pi^l_{\theta'}}(s) \geq V^h_{\pi^h_\phi,\pi^l_\theta}(s). \tag{22}$$

$\square$

### A.3 PROOF OF THEOREM 2

**Theorem 2.** *With the policy update formulae in Equations 5 and 6, the state value function $V^h_{\pi^h_{\phi_k},\pi^l_{\theta_k}}(s)$ will finally converge. $\pi^h_{\phi_k}, \pi^l_{\theta_k}$ are the $k$-th version of high-level policy and low-level policy respectively.*

*Proof.* Obviously, the $H$-timestep rewards have an upper bound $H \cdot r_{\max}$ where $r_{\max}$ is the maximum reward given by the original MDP. Therefore, there is a value upper bound in the TA-MDP:

$$\frac{H \cdot r_{\max}}{1-\tilde{\gamma}}. \tag{23}$$

Based on Theorem 1, we can know that $V^h_{\pi^h_{\phi_k},\pi^l_{\theta_k}}(s)$ increases monotonically. In summary, $V^h_{\pi^h_{\phi_k},\pi^l_{\theta_k}}(s)$ converges as $k \to \infty$. $\square$

### A.4 PROOF OF THEOREM 3

**Theorem 3.** *If the MDP $(\mathcal{S},\mathcal{A},p,r,\gamma)$ and the TA-MDP $(\mathcal{S},\mathcal{A},p_{\pi^l,H},\tilde{r},\tilde{\gamma})$ satisfy that $\tilde{\gamma} = \gamma^H$, then $\forall s \in \mathcal{S}, \tilde{\gamma}\cdot V^h_{\pi^h,\pi^l}(s) \leq V_{\pi^h,\pi^l}(s)$. It means that optimizing $V^h_{\pi^h,\pi^l}(s)$ is equivalent to optimizing a lower bound of $V_{\pi^h,\pi^l}(s)$.*

*Proof.* We can expand $\tilde{\gamma}\cdot V^h_{\pi^h,\pi^l}(s)$ with Equation 10 as follows:

$$\tilde{\gamma}\cdot V^h_{\pi^h,\pi^l}(s) = \tilde{\gamma}\cdot\int\sum_{\triangle t=0}^\infty \rho_{\triangle t}(s',a'|s,\pi^h,\pi^l)\tilde{\gamma}^{\lfloor\triangle t/H\rfloor}\cdot r(s',a')da'ds' \tag{24}$$

$$= \int\sum_{\triangle t=0}^\infty \rho_{\triangle t}(s',a'|s,\pi^h,\pi^l)\tilde{\gamma}^{\lfloor\triangle t/H\rfloor+1}\cdot r(s',a')da'ds'. \tag{25}$$

Because $r$ is always non-negative and the discount factor $\tilde{\gamma} \in (0,1]$, we know $\tilde{\gamma}\cdot V^h_{\pi^h,\pi^l}(s)$ satisfies that:

$$\leq \int\sum_{\triangle t=0}^\infty \rho_{\triangle t}(s',a'|s,\pi^h,\pi^l)\tilde{\gamma}^{\triangle t/H}\cdot r(s',a')da'ds'. \tag{26}$$

We can replace $\tilde{\gamma}$ with $\gamma^H$ and get:

$$= \int \sum_{\triangle t=0}^{\infty} \rho_{\triangle t}(s', a'|s, \pi^h, \pi^l)(\gamma^H)^{\triangle t/H} \cdot r(s', a')da'ds' \tag{27}$$

$$= \int \sum_{\triangle t=0}^{\infty} \rho_{\triangle t}(s', a'|s, \pi^h, \pi^l)\gamma^{\triangle t} \cdot r(s', a')da'ds' \tag{28}$$

$$= V_{\pi^h,\pi^l}(s). \tag{29}$$

In summary, $\forall s \in \mathcal{S}, \tilde{\gamma} \cdot V_{\pi^h,\pi^l}^h(s) \leq V_{\pi^h,\pi^l}(s)$, which means that optimizing $V_{\pi^h,\pi^l}^h(s)$ is equivalent to optimizing a lower bound of $V_{\pi^h,\pi^l}(s)$. $\qquad\square$

## B  VISUALIZATION OF SKILL SPACE COLLAPSE

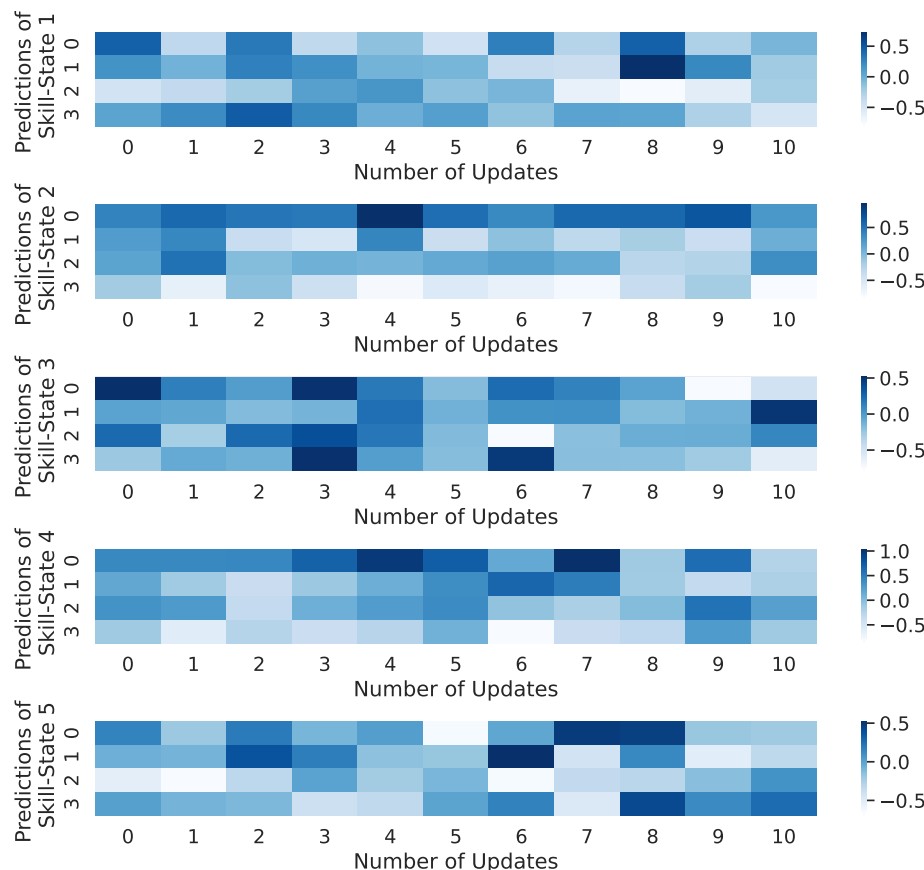

Figure 10: Each sub-figure represents the change process of actions predicted by the low-level policy for a state-skill. Each column of the sub-figure represents a predicted action, and the four rows in this column represent the four components of the action. The $i$-th column represents the predicted action after $i$ updates of the hierarchical policy. The predicted actions after two updates are significantly different from the initial predicted actions.

To visually show the process in which skill space collapse occurs, we visualize the change of the predicted actions given by the low-level policy for 5 state-skills that are not used for updating as the whole hierarchical policy is updated. This process of change takes place in the PyramidStack task adopted in Section 6 and is shown in Figure 10. We find that after 1 update, the predicted actions still remain highly similar to the initial predicted actions. However, after 2 and more updates, the

predicted actions and the initial predicted actions are obviously different, although the low-level policy is never updated with respect to these state-skills.

## C   ALGORITHM OF UPDATING HIGH-LEVEL POLICY AND SKILLS SIMULTANEOUSLY

---

**Algorithm 1** DSR: Dynamical Skill Refinement in Skill-based RL

---

**Input:** Pretrained high-level policy $\pi_{\text{off}}^h$, extracted low-level policy $\pi_{\text{off}}^l$, length of temporal abstraction $H$

**Output:** High-level policy $\pi_\phi^h$, residual low-level policy $\pi_\theta^l$, target network $\hat{f}$, variable network $f_\xi$

1: Initialize $\pi_\phi^h = \pi_{\text{off}}^h$ and create $\pi_\theta^l, \hat{f}, f_\xi$
2: **for** each epoch **do**
3:     Create high-level buffer $\beta^h = \emptyset$ and low-level buffer $\beta^l = \emptyset$
4:     **for** each episode **do**
5:         Initialize trajectory queue $tq = []$
6:         **for** timestep $t = 0, 1, 2, ...$ **do**
7:             **if** $t \mod H = 0$ **then**
8:                 Sample a skill $z \sim \pi_\phi^h(\cdot|s_t)$
9:             **end if**
10:            Decode an action $a_t \sim \pi_{\text{off}}^l(\cdot|s_t, z)$ and an action increment $\hat{a}_t \sim \pi_\theta^l(\cdot|s_t, z)$
11:            Map prediction error $||\hat{f}((s_t, z); \xi) - f((s, z))||_2$ to a weight $\alpha_t$
12:            Add weighted increment to action $a_t \leftarrow a_t + \alpha_t \cdot \hat{a}_t$
13:            Execute $a_t$ and get $r_t, s_{t+1}$
14:            Append $(s_t, z, \hat{a}_t, r_t, s_{t+1})$ to $tq$
15:        **end for**
16:        Extract high-level trajectory from $tq$ and store it:
             $\beta^h \leftarrow \beta^h \cup \{(..., (s_{\hat{t}\cdot H}, z_{\hat{t}\cdot H}, \hat{r}_{\hat{t}\cdot H}, s_{(\hat{t}+1)\cdot H}), ...)\}$
17:        Divide $tq$ into $H$-timestep rollouts and store them:
             $\beta^l \leftarrow \beta^l \cup \{..., ((s_{\hat{t}\cdot H}, z_{\hat{t}\cdot H}, \hat{a}_{\hat{t}\cdot H}, r_{\hat{t}\cdot H}, s_{\hat{t}\cdot H+1}), ...,$
             $(s_{\hat{t}\cdot H+H-1}, z_{\hat{t}\cdot H}, \hat{a}_{\hat{t}\cdot H+H-1}, r_{\hat{t}\cdot H+H-1}, s_{\hat{t}\cdot H+H})), ...\}$
             where each $r_{\hat{t}\cdot H+H-1}$ has been processed as Figure 3
18:    **end for**
19:    Apply PPO on $\beta^h$ to update $\pi_\phi^h$
20:    Apply PPO on $\beta^l$ to update $\pi_\theta^l$
21:    Minimize $||\hat{f}(\cdot; \xi) - f(\cdot)||_2$ with all $(s, z)$ in $\beta^l$ to update $\xi$
22: **end for**
23: **return** $\pi_\phi^h, \pi_\theta^l, \hat{f}, f_\xi$

---

In the online learning stage, the high-level policy and the skills interact with the environment to generate the trajectories which can be converted to the trajectories in the TA-MDP and divided into the constructed rollouts for refining the skills. We can directly use PPO Schulman et al. (2017) to finetune the high-level policy. With the constructed rollouts, the skills can also be directly refined through PPO. Notably, when the behavior of a skill $z$ in a state $s$ is refined, the predictor network's prediction error $||\hat{f}((s, z); \xi) - f((s, z))||_2$ regarding the state-skill $(s, z)$ should be accordingly optimized.

During the interaction between the hierarchical policy and the environment, when a skill $z$ predicts an action in the current state $s$, the residual policy predicts the action increment of the skill in the state. We convert the prediction error of the predictor network in this state-skill $(s, z)$ into a weight in the interval $[0, 1]$, assign this weight to the action increment, and add the weighted action increment to the predicted action to get the practical action. This conversion can be implemented by a scaled and shifted Sigmoid function.

The complete algorithm that incorporates dynamical skill refinement is detailedly illustrated in Algorithm 1. Obviously, the high-level policy and skills are updated only at the end of each epoch.

During the epoch, all the skills remain fixed, which means that the transition dynamics of the TA-MDP are stationary. Due to the natural circumvention of the temporal abstraction shift, we can directly use the PPO algorithm to update both the high-level policy and skills. The dynamical skill refinement mechanism is easily integrated into our algorithm. Its predictor network is updated alongside the skills, and the only way it influences the action is by assigning a dynamical weight to the action increment.

## D  DETAILS OF EXPERIMENTS

### D.1  DISCOUNT THE INNER-SKILL REWARDS

We find that the values of all the inner-skill state-actions are the same when the reward function is sparse, which is shown in Figure 3. However, in sparse-reward tasks, we want to solve the task as soon as possible; therefore, we propose to optimize the expected sum of the single-timestep rewards which are discounted by $\gamma = \tilde{\gamma}^{\frac{1}{H}}$ after every timestep rather than being discounted by $\tilde{\gamma}$ after every $H$ timesteps. We prove theoretically in Corollary 1 that such a skill refinement is equivalent to optimizing the lower bound of the performance in TA-MDP, which means that it is still unified with the high-level policy.

**Corollary 1.** *If the original MDP* $(\mathcal{S}, \mathcal{A}, p, r, \gamma)$ *and the TA-MDP* $(\mathcal{S}, \mathcal{A}, p_{\pi^l, H}, \tilde{r}, \tilde{\gamma})$ *satisfy that* $\tilde{\gamma} = \gamma^H$, *then* $\forall s \in \mathcal{S}, V_{\pi^h, \pi^l}(s) \leq V_{\pi^h, \pi^l}^h(s)$.

*Proof.* We can expand $V_{\pi^h, \pi^l}(s)$ with Equation 9 as follows:

$$V_{\pi^h, \pi^l}(s) = \int \sum_{\triangle t=0}^{\infty} \rho_{\triangle t}(s', a'|s, \pi^h, \pi^l)\gamma^{\triangle t} \cdot r(s', a')da'ds'. \tag{30}$$

Then, we rewrite it in the equivalent form:

$$= \int \sum_{\triangle t=0}^{\infty} \rho_{\triangle t}(s', a'|s, \pi^h, \pi^l)(\gamma^H)^{\triangle t/H} \cdot r(s', a')da'ds' \tag{31}$$

$$= \int \sum_{\triangle t=0}^{\infty} \rho_{\triangle t}(s', a'|s, \pi^h, \pi^l)\tilde{\gamma}^{\triangle t/H} \cdot r(s', a')da'ds'. \tag{32}$$

We enlarge it and get:

$$\leq \int \sum_{\triangle t=0}^{\infty} \rho_{\triangle t}(s', a'|s, \pi^h, \pi^l)\tilde{\gamma}^{\lfloor \triangle t/H \rfloor} \cdot r(s', a')da'ds' \tag{33}$$

$$= V_{\pi^h, \pi^l}^h(s). \tag{34}$$

In summary, if we refine the skills with respect to the expected sum of the single-timestep rewards discounted by $\tilde{\gamma}^{\frac{1}{H}}$, we actually optimize a lower bound of the state-value in the TA-MDP. $\qquad\square$

Intuitively, we can consider such a skill refinement as optimizing the low-level policy $\pi^l$ in an MDP $(\hat{\mathcal{S}}, \hat{\mathcal{A}}, \hat{p}_{\pi^h}, \hat{r}, \hat{\gamma})$. The state space $\hat{\mathcal{S}} = \mathcal{S} \times \mathcal{Z}$ is Cartesian product of the state space $\mathcal{S}$ of the original MDP and the skill space $\mathcal{Z}$. The action space $\hat{\mathcal{A}} = \mathcal{A}$ is equal to the original action space. The transition dynamics $\hat{p}_{\pi^h}$ is determined by the original transition dynamics $p$ and the high-level policy $\pi^h$. The reward function $\hat{r} : \mathcal{S} \times \mathcal{A} \to \mathbb{R}$ maps the original state component and the action to the reward value which the original reward function will also give. The discount factor $\hat{\gamma} = \tilde{\gamma}^{\frac{1}{H}}$. The trajectory generated by the interaction of the entire hierarchical policy with the environment can be viewed as the rollout of the low-level policy. From this perspective, we can directly use the PPO Schulman et al. (2017) algorithm to optimize the low-level policy. We find in experiments that this practical implementation of skill refinement works well.

## D.2 DETAILS OF DATASETS AND DOWNSTREAM TASKS

We reuse the datasets from ReSkill Rana et al. (2023), which were collected by hand-scripted controllers. The fetch_block_40000 dataset consists of 40,000 trajectories collected from three tasks: TableCleanup, SlipperyPush and PyramidStack. We extract the skills from it for the physically modified versions of these tasks which serve as downstream tasks. The fetch_hook_40000 dataset consists of 40,000 trajectories collected from ComplexHook. We extract the skills from it for the physically modified version of ComplexHook which serves as the downstream task.

When these tasks are used as downstream tasks, their physical properties are modified appropriately in order to make skill refinement more necessary. In SlipperyPush, the friction of the table surface is reduced compared to the transitions in the dataset. In TableCleanup, the tray is actually added and the agent has to overcome the edges of the tray which are the obstacles. In PyramidStack, the gripper can reach higher heights. In ComplexHook, the objects are drawn randomly from a library of unseen objects and the table surface is scattered rigid obstacles. The lengths of episodes in TableCleanup, PyramidStack, SlipperyPush and ComplexHook are 50, 50, 100 and 100 respectively. These details remain consistent with ReSkill.

## D.3 MAP PREDICTION ERROR TO THE WEIGHT OF ACTION INCREMENT

| Downstream Task | Hyper-parameter | Value |
|---|---|---|
| TableCleanup | $\alpha$ | 1 |
| TableCleanup | $k$ | -300 |
| TableCleanup | $c$ | 0.025 |
| SlipperyPush | $\alpha$ | 1 |
| SlipperyPush | $k$ | -300 |
| SlipperyPush | $c$ | 0.025 |
| PyramidStack | $\alpha$ | 0.6 |
| PyramidStack | $k$ | -10 |
| PyramidStack | $c$ | 0.01 |
| ComplexHook | $\alpha$ | 1 |
| ComplexHook | $k$ | -20 |
| ComplexHook | $c$ | 0.04 |

Table 1: Hyper-parameters of mapping prediction error to weight of action increment.

We point out that the prediction error given by distillation networks can be mapped to the weight of the action increment by the following scaled and shifted Sigmoid function.

$$S(x) = \alpha \frac{1}{1 + e^{-k(x-c)}}. \tag{35}$$

$\alpha, k, c$ can be seen as the hand-drafted task-specific hyper-parameters. The values of these hyper-parameters in different downstream tasks are shown in Table 1. We can collect very few transitions generated by agent's interacting with downstream tasks and visualize the prediction error curves to determine appropriate values of these hyper-parameters. The architectures of the prediction and target networks are the same and illustrated in Table 2.

| Properties | Value |
|---|---|
| Hidden sizes | $[64, 64]$ |
| Activation function | Tanh |
| Output activation function | Identity |
| Optimizer | Adam |
| Learning rate | $3e-4$ |

Table 2: Architecture of the prediction and the target networks.

### D.4 DETERMINE $\alpha, k, c$ WITH A FEW OF ONLINE INTERACTIONS

We use the TableCleanup task as an example to show how to determine the hyper-parameters $\alpha, k, c$ used to map the prediction error to the weight of the action increment. We use a few of online interactions to select the appropriate values of $\alpha, k, c$. We use only 50 epochs to tentatively train the hierarchical policy and variable prediction network and record the prediction errors. We visualize these prediction errors in Figure 11. We find that the prediction errors concentrate below $0.025$ as the training progresses. Before that, the prediction errors hover in $(0.025, 0.04]$. When the prediction errors are above $0.04$, they show a trend of divergence. It is natural to consider the state-skill with a prediction error less than $0.025$ as a sufficiently refined state-skill. This observation motivates us to set $c$ to $0.025$, which means that when the prediction error is around $0.025$, the mapped weight increases rapidly. When the prediction error approaches $0$, we expect the weight to approach $1$. With this goal, we can simply set $\alpha$ equal to $1$ and just adjust $k$. Since the adjustment process is done on the collected data, it involves no online interaction costs. The mapping from prediction error to weight obtained by adjusting $k$ is presented in Figure 12.

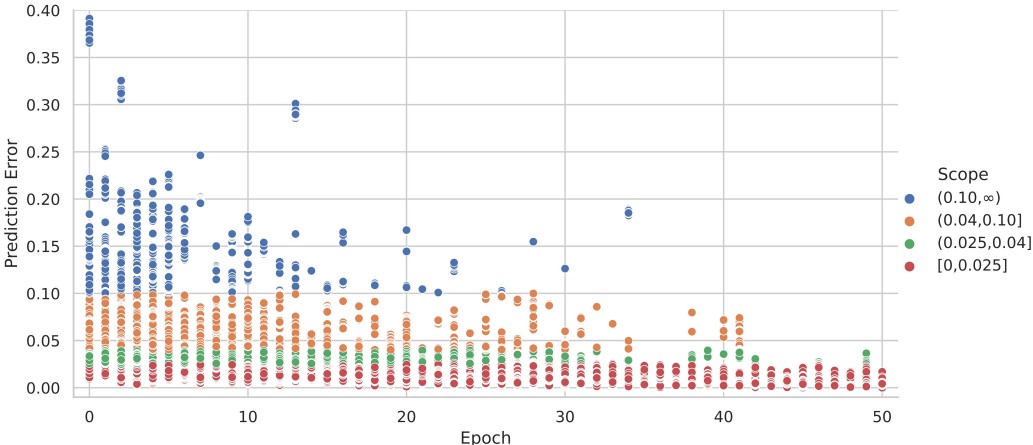

Figure 11: The prediction errors are given different colors according to the scopes to which they belong.

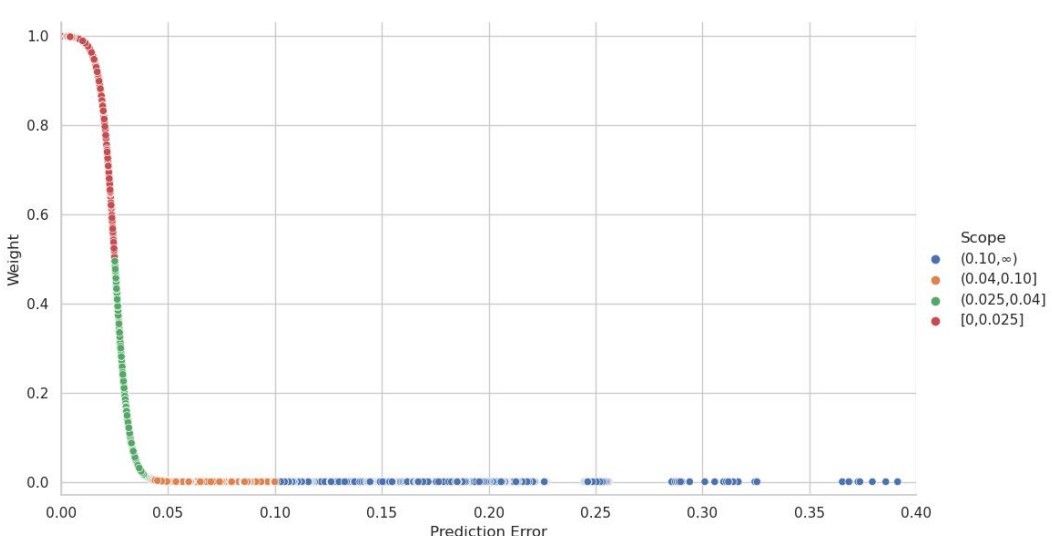

Figure 12: After adjusting, by setting $\alpha = 1$ and $c = 0.025$, we find that $k = -300$ works as expected.

