# OpenReview forum: "DSR: Reinforcement Learning with Dynamical Skill Refinement"
_ICLR.cc/2025/Conference — Submitted to ICLR 2025_

### Official Review · Reviewer_tpGY · 2024-10-26

**Soundness:** 2
**Presentation:** 2
**Contribution:** 3
**Rating:** 5
**Confidence:** 4

**Summary:**

This paper addresses the problem of the sub-optimal dataset in the skill-based reinforcement learning settings. The authors propose a unified optimization objective to train a hierarchical skill structure, providing a mathematical proof within the TA-MDP. Additionally, the authors introduce a dynamic skill refinement approach to mitigate potential collapse in the skill space.

**Strengths:**

- Learning skills from sub-optimal datasets and overcoming skill space collapse are crucial challenges in skill-based reinforcement learning scenarios.

**Weaknesses:**

- I am not fully convinced how the unified optimization objective addresses the issue of sub-optimality in the datasets. DSR learns the low-level policy, making it more likely to optimize the action sequence for the downstream policy. If this is the case, the authors should emphasize more on mitigating the collapse in the skill space.
- The contribution on dynamic skill refinement (Section 5.2) seems minimal, and its benefits are not fully demonstrated in Ablation 6.2, as DirectRefinement also shows comparable performance. Otherwise, the authors should provide more ablation studies on the learning of TA-MDP.
- In Section 6.1, the authors mention that DSR improves performance by avoiding skill space collapse. However, in Ablation 6.2, skill space collapse appears to occur only in the PyramidStack task. If the authors want to emphasize their contributions regarding skill space collapse, they should create more scenarios similar to PyramidStack and elaborate on why these tasks cause the collapse, and why it does not occur in other tasks.
- The presentation could be further improved. Figure 4 is smaller compared to Figure 2, and Figure 3 could be replaced with an equation for better clarity. For clarity, it would be better to include the pseudocode in the main manuscript rather than in Appendix C.

**Questions:**

- As the paper addresses the problem arising from the sub-optimality (low quality and low coverage) of the datasets, the authors should elaborate more on their dataset collection regarding to its sub-optimality.
-  In Figure 6, the y-axis denotes the episodic return. How is the episodic return computed? Additionally, can the authors elaborate more on their sparse reward setting? Is a reward given only upon task completion?
- In Ablation 6.2, what do the authors mean by "the extracted skills are not well initialized"? In Appendix D.2, the authors mention using datasets collected from TableCleanup, SlipperyPush, and PyramidStack, but why does only the PyramidStack task show this performance gap in Ablation 6.2?
- According to the original ReSkill paper, ReSkill achieves an average return of over 15 in the PyramidStack task. However, in Figure 6, ReSkill achieves only an average return of 10. The authors mentioned that the experimental settings are equivalent to those in ReSkill as described in Appendix D.2. Why is there a performance gap?
- It would be benificial, if the authors provide more ablation studies on their learning mechanism of TA-MDP. (Section 4)
- Can the authors provide additional experiments on other robotic manipulation environments, including long-horizon scenarios such as MetaWorld or Franka Kitchen?

---

> ### Author Response · Authors · 2024-11-17
> **Dear reviewer, the following are responses to your questions**
>
> **About Weakness**
> * **Weakness 1** The key to solving the sub-optimality of the extracted skills is to ensure performance improvement during online learning. There is a lack of theoretical foundation for previous works of Skill-based RL with skill refinement. We certainly need to first address the issue of how to ensure performance improvement. I think you haven't figured out the root cause of skill space collapse is that all the skills are implemented through the same parametric model, which is the second problem. If you can propose a method to optimize low-level policies regarding a given skill while keeping the behaviors of other skills unchanged, then this unified optimization objective alone can achieve simultaneous optimization of high-level policies and skills.
>
> * **Weakness 2** Obviously, the comparable performance between DirectRefinement and DSR shown in Figure 7 precisely demonstrates that DSR avoids performance crashes caused by skill space collapse without sacrificing performance improvement speed, which is rare for conservative based measures. Usually, conservative measures will slow down the speed of performance improvement when the measures are unnecessary. In PyramidStack, conservative measures are necessary, and DSR avoids performance crashes. In the other three tasks, conservatism is not necessary, and DSR did not show a slower performance improvement than DirectRefinement. Yet, we will try to add ablation studies in latter versions of this paper.
>
> * **Weakness 3** The primary focus of this paper is the optimization objective which ensures the performance improvement, followed by implementing skill refinement with the dynamical skill refinement mechanism. Perhaps we should change the title to emphasize this point, for example: "Dynamically Refine Skills under Unified Optimization Objective". We believe that we should not deliberately create situations where skill space collapse occurs, as the performance of each task is averaged across multiple random seeds. Therefore, we believe that whether performance crashes will occur is task related.
>
> * **Weakness 4** We will consider further improving the presentation in the latter versions.
>
> **Answers to Questions**
> * **Question 1** Since the benchmark used in this paper is from ReSkill, we have adopted ReSkill's dataset which is generated through hand-scripted controllers. The physical properties of downstream tasks have been modified, making them different from the tasks used to generate the dataset, thus creating sub-optimality of the dataset.
>
> * **Question 2** Yes, a non-zero reward is given only when the task is completed, but the earlier the task is completed, the greater the reward will be.
>
> * **Question 3** "The extracted skills are not well initialized" means that the skills extracted from the dataset can not be used to effectively solve a downstream task, which is reflected in the difficulty of improving the performance curve. I think this performance gap is just task-related, as all performance is averaged over four random seeds.
>
>
> * **Question 4** I assure you very clearly that we downloaded the codes provided by the authors of ReSkill and reproduced the experiments using the codes, and the result is what we provided in this paper. **I think it may be that the publicly available version of ReSkill's tasks is different from the version they actually used. We tried to contact them, but did not receive a response.**
>
> * **Question 5 and Question 6** We will try to add experimental analysis during the rebuttal period, and if we can complete it, we will add corresponding experimental analysis in latter versions of this paper.

---

> > ### Comment · Reviewer_tpGY · 2024-11-22
> >
> > I appreciate the authors for addressing my previous questions. Based on their response, most of my questions regarding the contribution of the work and the experimental settings have been resolved. Thus, I raise my score to 5.
> >
> > However, I still have questions about the method's robustness across diverse environments where skill collapse might occur, as DSR only shows its benefit (in resolving skill collapse) solely in the Pyramid stack task. To convincingly demonstrate the benefits of the proposed approach, I think that it is crucial to show its performance in scenarios where skill collapse is more prevalent.

---

> ### Author Response · Authors · 2024-12-03
> **Dear reviewer, we have submitted the revised version**
>
> # Dear reviewer:
>    * **In the revised version of this paper, we have made the following modifications:**
>
> ## Experiment
> * We add the ablation analysis of the hyper-parameters involved in dynamical skill refinement and show it in Figure 9.
> * This analysis demonstrates the insensitivity of our method to the hyper-parameters.
>
> ## Title
> * In order to help readers better understand the contribution of this paper, we have revised the title to: "DSR: Optimization of Performance Lower Bound for Hierarchical Policy with Dynamical Skill Refinement"
> * Our first contribution is to provide a theoretical explanation for the effectiveness of skill-based RL, that is, optimizing the entire hierarchical policy in TA-MDP can be equivalent to optimizing its performance lower bound in the original MDP. We believe the new title can reflect this.
>
>
> ## Figure
> * We improved Figures 1, 2, 3, 4, and 8.
> * The font and size in Figure 4 have been improved to present the content more clearly.
>
> ## Notations
> * We found several notation mistakes, where $\pi^l_{\theta}$ was written as $\pi^h_{\theta}$. We apologize for this and have already made the necessary corrections.
>
> ## Reference
> * We have added citations to two closely related papers:
>   2. Nachum, Ofir, et al. "Near-optimal representation learning for hierarchical reinforcement learning." arXiv preprint arXiv:1810.01257 (2018).
>   3. Yang, Yiqin, et al. "Flow to control: Offline reinforcement learning with lossless primitive discovery." Proceedings of the AAAI Conference on Artificial Intelligence. Vol. 37. No. 9. 2023.
>
> ## Correction
> * We found that the hyper-parameters described in Appendix D.4 were incorrect and have made corrections to address this issue.

---

### Official Review · Reviewer_uuWW · 2024-11-02

**Soundness:** 3
**Presentation:** 2
**Contribution:** 2
**Rating:** 6
**Confidence:** 3

**Summary:**

This paper aims to address two issues in RL with skills: **(1)** sub-optimal pretrained skills due to low quality and low coverage of the datasets, and **(2)** temporabl abstraction shift due to refining skills. The authors propose to unify the optimization objectives of both high-level and low-level policies as the future return and use random network distillation to dynamically adjust the weight of action refinement. The paper demonstrates the effectiveness of the proposed method by providing theoretical analysis and empirical results.

**Strengths:**

* The proposed method is well-motivated, theoretically grounded, and demonstrates better performance against several state-of-the-art RL-with-skills methods.
* The author uses multiple figures to improve the clarity of the paper.

**Weaknesses:**

While well-motivated, I have some questions about this work:

1. In the introduction, the authors mention that previous works (e.g., SPiRL [1]) assumes near-optimal dataset to pretrain the skills. However, even if the skill contains some sub-optimal behavors, could the high-level policy still learns to avoid choosing the bad skills as it aims to maximize the reward?
2. From the theoretical perspective, how is DSR different from the classic online hierarchical RL methods? For instance, [2] updates both high-level and low-level policies with RL objective with representation learning, and also provides the optimization objective bound. A similar bound is provided in [3] for offline settings. These two bounds actually quantify the exact form of sub-optimality, hence I think they are stronger than Theorem 3 in this work. I would appreciate it if the authors could discuss these methods in the paper.
3. How well does DSR can transfer skills to new tasks? I think one promise of skill-based RL is that it can conbine the learned skills to novel tasks, as mentioned in [1]. I am wondering if the authors could show some results in their experimental setting, such as training on data from one task and evaluating on other tasks.

Overall, while this paper presents some interesting ideas, I am unable to recommend acceptance at this stage given the questions mentioned above. However, I would consider raising my score if the authors could address my concerns.

[1] Pertsch, Karl, Youngwoon Lee, and Joseph Lim. "Accelerating reinforcement learning with learned skill priors." Conference on robot learning. PMLR, 2021.

[2] Nachum, Ofir, et al. "Near-optimal representation learning for hierarchical reinforcement learning." *arXiv preprint arXiv:1810.01257* (2018).

[3] Yang, Yiqin, et al. "Flow to control: Offline reinforcement learning with lossless primitive discovery." Proceedings of the AAAI Conference on Artificial Intelligence. Vol. 37. No. 9. 2023.

**Questions:**

There are some questions and concerns, which I have outlined in the previous section.

---

> ### Author Response · Authors · 2024-11-17
> **Dear reviewer, the following are responses to your questions**
>
> **Responses to the weakness**
> * **Weakness 1** Including sub-optimal behaviors is not the fundamental issue, as we can still find the optimal skills through explorations. The fundamental problem is that there are no optimal behaviors for downstream tasks in the extracted skills, which is because in Skill-based RL, the dataset used to extract skills is usually task-agnostic, so it may not necessarily include high-quality demonstrations for the downstream task. This makes it impractical to assume that the dataset is near-optimal. It becomes necessary to refine the extracted skills from the dataset.
>
> * **Weakness 2** We confirm that we have read these two cited references [2,3] before.
>   * **Weakness 2-Part 1** Reference [2] addresses how to map states to the goal space. Essentially, they are optimizing the mapping relationship between goal space $G$ and policy space $\Pi$, which can be seen as a measure to enhance exploration efficiency. Policy space $\pi(a|s_{t+k},k)\in\Pi$ is assumed to be sufficient for downstream tasks. But our paper essentially addresses how to optimize the behaviors of the policies in policy space.
>   * **Weakness 2-Part 2** Although the boundaries in reference [3] seem more complex, they explore the difference between the performance of offline extracted hierarchical policy from a fixed dataset using the specific PEVI method and the performance of the optimal policy. Additionally, this bound assumes the measurable size $|\Pi_\theta|$ of policies, the existence of a finite concentration coeffcient $c^\dagger$ and so on. The bound in our paper is used for both refining skills and learning high-level policy during online learning stage, and is actually the optimization objectives. The significance of our bound is to provide an optimized lower bound that can be used to continuously improve the performance during the online learning stage.
>   * Yet, we still believe that is meaningful to reference these two papers. We promise to add them to the related work in the latter version of our paper.
>
>
> * **Weakness 3** The experiments in this paper has already transferred the skills to the new tasks. We mentioned in Appendix D.2 that the downstream tasks are physically modified. Although they look the same, the physical properties of downstream tasks have been modified, making skill refinements necessary, which is consistent with ReSkill because we want to better compare with previous works.

---

> > ### Comment · Reviewer_uuWW · 2024-11-24
> > **Thank you for the response**
> >
> > Thank the authors for the response. After reading the response and other reviewers' comments, I would like to maintain my score.

---

> ### Author Response · Authors · 2024-12-03
> **Dear reviewer, we have submitted the revised version**
>
> # Dear reviewer:
>    * **In the revised version of this paper, we have made the following modifications:**
>
> ## Experiment
> * We add the ablation analysis of the hyper-parameters involved in dynamical skill refinement and show it in Figure 9.
> * This analysis demonstrates the insensitivity of our method to the hyper-parameters.
>
> ## Title
> * In order to help readers better understand the contribution of this paper, we have revised the title to: "DSR: Optimization of Performance Lower Bound for Hierarchical Policy with Dynamical Skill Refinement"
> * Our first contribution is to provide a theoretical explanation for the effectiveness of skill-based RL, that is, optimizing the entire hierarchical policy in TA-MDP can be equivalent to optimizing its performance lower bound in the original MDP. We believe the new title can reflect this.
>
>
> ## Figure
> * We improved Figures 1, 2, 3, 4, and 8.
> * The font and size in Figure 4 have been improved to present the content more clearly.
>
> ## Notations
> * We found several notation mistakes, where $\pi^l_{\theta}$ was written as $\pi^h_{\theta}$. We apologize for this and have already made the necessary corrections.
>
> ## Reference
> * We have added citations to two closely related papers:
>   2. Nachum, Ofir, et al. "Near-optimal representation learning for hierarchical reinforcement learning." arXiv preprint arXiv:1810.01257 (2018).
>   3. Yang, Yiqin, et al. "Flow to control: Offline reinforcement learning with lossless primitive discovery." Proceedings of the AAAI Conference on Artificial Intelligence. Vol. 37. No. 9. 2023.
>
> ## Correction
> * We found that the hyper-parameters described in Appendix D.4 were incorrect and have made corrections to address this issue.

---

### Official Review · Reviewer_qK82 · 2024-11-04

**Soundness:** 3
**Presentation:** 3
**Contribution:** 2
**Rating:** 5
**Confidence:** 3

**Summary:**

The paper introduces Dynamical Skill Refinement (DSR), an on-policy reinforcement learning method designed to optimize hierarchical policies in environments with sparse rewards. The paper refines the skills in an on-policy manner and proves that the temporal abstraction shift is circumvented by simultaneously updating the high-level policy and skills. The method prevents skill space collapse, which can lead to performance issues, by incorporating a dynamical mechanism that refines skills without disrupting the latent space. Empirical results demonstrate that DSR outperforms state-of-the-art methods in complex sparse-reward robotic manipulation tasks.

**Strengths:**

1- The paper is well-written and easy to read.
2- The proofs for theorems are solid.
3- Good comparisons with SOTA methods.

**Weaknesses:**

1- The main difference between your approach and ReSKILL is that yours handles the temporal abstraction shift but it is not shown in your theorems.
2- This method needs lots of extra parts such as an extra residual policy to avoid skill collapse. Also, using RND makes sure that a skill is refined enough but still you cannot make sure that modifying the skill does not impact the state of other skills in the same state. Moreover, RND is an approach that needs lots of tuning such as how you control the learning rate of the variable network and the threshold for being accurate enough for the refinement. This would impact the results and we expected to see an analysis of these parameters.

3- Usually we have complex navigation robotic environments in the experiments with skills as the macro-action to reach the subgoals or landmarks. We do not see any navigation environment here.

**Questions:**

1- Can you explain again how skills are extracted using VAE?
2- ReSKILL has a similar on-policy approach as your method. You said that ReSKILL cannot theoretically guarantee performance improvement, but you did not mention proof. Can you elaborate more on this?

---

> ### Author Response · Authors · 2024-11-14
> **Dear reviewer, the following are responses to your questions**
>
> **About the Weakness**
> * **Weakness 1** The solution to temporal abstraction shift is to update both the high-level policy and the low-level policy in an on-policy RL manner, rather than the theorems mentioned in this paper. The reason why we propose these theorems is that previous works simply learn the hierarchical policy in the TA-MDP, without theoretically analyzing the effectiveness. Namely, our theorems answer the question why skill-based RL works. Combining the on-policy RL manner with our theorems can ensure the performance improvement in the original MDP while avoiding temporal abstraction shift.
> * **Weakness 2** In this paper, the input of RND is state-skill, instead of state only. Therefore, a smaller RND prediction error will let only a skill's behavior in the given state be refined to a greater extent. In the experiment, we found that only the parameters of the mapping function that maps RND errors to the weight of action increments require special tuning. We provide an operation example in Appendix D.4 to determine the appropriate hyper-parameters of the mapping function using a small number of online experiences.
> * **Weakness 3** We believe that the essential difference between navigation tasks and robotics is that in navigation tasks, skill space is usually represented by the state space. In this case, skill-based RL will be transformed into goal-conditional RL. Similarly, ReSkill has also been validated totally in robotics tasks. **We will try to add new experiments during the rebuttal period, and if time permits, we will resubmit a new version of the paper.**
>
> **Answers to Questions**
> * **Question 1** We repeatedly sample fixed-length state-action segments from the trajectories in the dataset. Subsequently, we use an LSTM to map the state-action segments into feature vectors. Encoder maps feature vectors to the embedding space. Finally, decoder conditioned on the embedding and the states attempts to reconstruct the actions in the segments. The reconstruction error and regularization term will be used for joint training of the entire VAE and LSTM.
> * **Question 2** ReSkill, like previous works (e.g. [1,2,3]), learns high-level policies in TA-MDP without explaining why this is effective. Because the original task of maximizing the expected return in the original MDP has been transformed into maximizing the expected return in TA-MDP. ReSkill even learns high-level policy in TA-MDP and low-level policy in Original MDP respectively, which exacerbates the interpretability problem of skill-based RL.
>   * **We novelly prove that optimizing the expected return in TA-MDP is equivalent to optimizing the expected return of the entire hierarchical policy in the original MDP, which is shown in Theorem 3. ($\forall s\in\mathcal{S},\tilde{\gamma}\cdot V^h_{\pi^h,\pi^l}(s)\leq V_{\pi^h,\pi^l}(s)$)**
>   * We have presented a optimization objective of refining the low-level policy in Figure 3 and demonstrated in Theorem 1 that this objective can ensure the performance improvement in TA-MDP. ($\forall s\in\mathcal{S},V^h_{\pi^h_{\phi'},\pi^l_{\theta'}}(s)\geq V^h_{\pi^h_{\phi},\pi^l_{\theta}}(s)$)
>   * Combining Theorem 2 and Theorem 3, it is not difficult to find that under this unified optimization objective, it is equivalent to optimizing a lower bound of the performance of the entire hierarchical policy in the original MDP. This provides an unprecedented theoretical foundation for skill-based RL, and the key to all of this is that the discount factor $\tilde{\gamma}$ in TA-MDP and the discount factor $\gamma$ in the original MDP satisfy the following condition: $\tilde{\gamma}=\gamma^H$.
>   * The solid theoretical arguments mentioned above are not present in ReSkill and [1,2,3]. However, since the discount factor $\gamma$ is usually set to a number close to 1, such as 0.99, its H-power $\gamma^H$ is close to itself, which is why those works are also effective in the experiments. **We believe that in the field of skill-based RL, the theoretical foundation lags far behind experiments, and as increasingly complex environments are adopted, we need to first address the key issue why it is effective.**
>
>
> **References**
>
> [1]. Karl Pertsch, Youngwoon Lee, and Joseph Lim. Accelerating reinforcement learning with learned skill priors. In Conference on robot learning, pp. 188–204. PMLR, 2021.
>
> [2]. Karl Pertsch, Youngwoon Lee, Yue Wu, and Joseph J Lim. Guided reinforcement learning with learned skills. In Conference on Robot Learning, pp. 729–739. PMLR, 2022.
>
> [3]. Ce Hao, Catherine Weaver, Chen Tang, Kenta Kawamoto, Masayoshi Tomizuka, and Wei Zhan. Skill-critic: Refining learned skills for hierarchical reinforcement learning. IEEE Robotics and Automation Letters, 2024.

---

> ### Author Response · Authors · 2024-11-14
> **May I ask why you changed your rating?**
>
> Dear reviewer:
> ﻿
> I am confused as to why you suddenly made significant changes to the rating? I plan to further update the content of the rebuttal and the paper. May I get your advice?
> ﻿
> thank you.

---

> > ### Comment · Reviewer_qK82 · 2024-11-14
> > **Change in my rating**
> >
> > Dear author,
> > Thanks for replying to my comments.
> >
> > I was uncertain about my score. After reading other reviewers' comments and noticing their confidence in their ratings, I decided to lower my score.

---

> ### Author Response · Authors · 2024-12-03
> **Dear reviewer, we have submitted the revised version**
>
> # Dear reviewer:
>    * **In the revised version of this paper, we have made the following modifications:**
>
> ## Experiment
> * We add the ablation analysis of the hyper-parameters involved in dynamical skill refinement and show it in Figure 9.
> * This analysis demonstrates the insensitivity of our method to the hyper-parameters.
>
> ## Title
> * In order to help readers better understand the contribution of this paper, we have revised the title to: "DSR: Optimization of Performance Lower Bound for Hierarchical Policy with Dynamical Skill Refinement"
> * Our first contribution is to provide a theoretical explanation for the effectiveness of skill-based RL, that is, optimizing the entire hierarchical policy in TA-MDP can be equivalent to optimizing its performance lower bound in the original MDP. We believe the new title can reflect this.
>
>
> ## Figure
> * We improved Figures 1, 2, 3, 4, and 8.
> * The font and size in Figure 4 have been improved to present the content more clearly.
>
> ## Notations
> * We found several notation mistakes, where $\pi^l_{\theta}$ was written as $\pi^h_{\theta}$. We apologize for this and have already made the necessary corrections.
>
> ## Reference
> * We have added citations to two closely related papers:
>   2. Nachum, Ofir, et al. "Near-optimal representation learning for hierarchical reinforcement learning." arXiv preprint arXiv:1810.01257 (2018).
>   3. Yang, Yiqin, et al. "Flow to control: Offline reinforcement learning with lossless primitive discovery." Proceedings of the AAAI Conference on Artificial Intelligence. Vol. 37. No. 9. 2023.
>
> ## Correction
> * We found that the hyper-parameters described in Appendix D.4 were incorrect and have made corrections to address this issue.

---

### Meta-Review · Area_Chair_yUki · 2024-12-20

**Metareview:**

This paper proposes a method for learning skills from a sub-optimal dataset, which are used in a hierarchical policy to solve sparse-reward tasks. A the key component of the method is a mechanism to prevent skill space collapse. The paper includes both theoretical analysis and empirical results demonstrating improved performance relative to state-of-the-art methods.

Reviewers appreciated the importance of the problem setting, the proposed method's strong empirical performance, the correctness of the theoretical results, and clear writing and motivation in the text.

Reviewers had concerns about the similarities with prior work (ReSKILL, SPiRL, classical HRL methods), including the novelty of Theorem 3 (which may have appeared in prior work). They also had concerns about the complexity of the proposed approach. Reviewers suggested additional baselines (ReSKILL) and evaluation tasks (complex navigation robotic environments, transfer to new settings, clarifying the results from Ablation 6.2), and minor presentation recommendations.

Overall, I think this has the makings of the strong paper (combining theoretical analysis with a new method), but there remain some unaddressed concerns about the relationship with prior work. I therefore am recommending that the paper be rejected. I would encourage the authors to resubmit after addressing the the feedback mentioned in the reviews.

**Additional Comments On Reviewer Discussion:**

During the rebuttal, the authors added new ablation experiments, revised the paper title. However, one reviewer still had concerns about whether the method's robustness.

---

### Decision · Program_Chairs · 2025-01-22

Reject